# Deep Learning-Emerged Grid Cells-Based Bio-Inspired Navigation in Robotics

**DOI:** 10.3390/s25051576

**Published:** 2025-03-04

**Authors:** Arturs Simkuns, Rodions Saltanovs, Maksims Ivanovs, Roberts Kadikis

**Affiliations:** Institute of Electronics and Computer Science, 14 Dzerbenes St., LV-1006 Riga, Latvia; saltanov.rodion@edi.lv (R.S.); maksims.ivanovs@edi.lv (M.I.); roberts.kadikis@edi.lv (R.K.)

**Keywords:** grid cells, deep learning, robotics, robotic navigation, entorhinal cortex, recurrent neural networks, mobile robots, path integration, spatial cognition, machine learning, autonomous systems

## Abstract

Grid cells in the brain’s entorhinal cortex are essential for spatial navigation and have inspired advancements in robotic navigation systems. This paper first provides an overview of recent research on grid cell-based navigation in robotics, focusing on deep learning models and algorithms capable of handling uncertainty and dynamic environments. We then present experimental results where a grid cell network was trained using trajectories from a mobile unmanned ground vehicle (UGV) robot. After training, the network’s units exhibited spatially periodic and hexagonal activation patterns characteristic of biological grid cells, as well as responses resembling border cells and head-direction cells. These findings demonstrate that grid cell networks can effectively learn spatial representations from robot trajectories, providing a foundation for developing advanced navigation algorithms for mobile robots. We conclude by discussing current challenges and future research directions in this field.

## 1. Introduction

Effective navigation in complex environments is a fundamental challenge in both neuroscience and robotics. In neuroscience, the discovery of grid cells in the entorhinal cortex has significantly advanced our understanding of spatial representation and navigation in mammals [1]. Grid cells fire in a repeating hexagonal pattern [2] that covers the entire navigable space, providing an intrinsic coordinate system for spatial mapping and path integration [3]. This hexagonal firing pattern allows for precise encoding of an organism’s position in space, independent of external cues [4].

In robotics, achieving robust and adaptable navigation remains a critical goal, especially in dynamic and unstructured environments. Traditional navigation algorithms often rely on external references or predefined maps, which can limit their effectiveness when encountering changes or uncertainties in the environment. Inspired by the biological mechanisms of spatial navigation, researchers have begun to explore the integration of grid cell models into robotic systems [5,6]. These models hold the potential to provide robots with an internal spatial representation that enables efficient path planning and navigation without reliance on external positioning systems.

Despite these advances, gaps remain in demonstrating how grid cell networks can be trained using real-world robotic data and how they can handle the complexities of physical environments. Existing studies [5] have primarily focused on simulated environments which datasets are generated by mathematical methods and graphical simulation environment without physics and without ROS intermediary communication layers or have not fully leveraged deep learning techniques to model grid cell representations in robots [7].

This paper addresses this gap by presenting both an overview of recent research on grid cell-based navigation in robotics and experimental results from training a grid cell network using trajectories collected from a mobile unmanned ground vehicle (UGV) robot. Specifically, we trained a recurrent neural network (RNN) utilizing the robot’s trajectories, resulting in the emergence of representations resembling grid cells and other cell types found in the entorhinal region, such as head-direction cells [8] and place cells [9]. Our findings support neuroscientific theories that consider grid cells essential for vector-based navigation [10] and the hypothesis that grid cells serve as a metric for spatial representation [11].

By demonstrating the emergence of grid cell-like patterns in artificial neural networks trained on actual robot trajectory data as illustrated in Figure 1, this work contributes to the development of more robust and adaptable navigation algorithms for mobile robots. It bridges the gap between neuroscientific models of spatial representation and practical robotic applications, providing a foundation for future research in this field.

Most prior grid-cell–inspired robotics research [5] relies on simplified, 2D simulations or 3D graphical environments (i.e., DeepMind Lab) that lack realistic physics or sensor/motor delays. By contrast, we use a ROS + Gazebo simulation environment that models collision dynamics, sensor noise, and communication latencies more faithfully. This realism improves generalization and robustness when transferring to real-world scenarios. We also randomize the robot’s starting position and orientation, exposing the model to a broad range of conditions and making it better equipped to handle unpredictable environments.

Furthermore, we introduce a new training pipeline that generates near-real Husky UGV trajectories—rather than purely synthetic or mathematically derived paths—via our ROS + Gazebo dataset module. These physics-based trajectories are richer and more varied, capturing the true motion constraints and sensor feedback typical of physical robots. Consequently, the learned representations and navigation behaviors are more adaptive and robust compared to approaches trained solely on highly idealized data.

## 2. Grid Cells in Mammals Cerebrum

Grid cells, discovered by May-Britt Moser and Edvard I. Moser in 2005 in the medial entorhinal cortex (MEC) of rats, are neurons important for spatial navigation and memory [1]. Initially identified in rats, grid cells have since been observed in other mammals, including humans. These cells form a hexagonal grid of activity, creating a coordinate system that allows animals to map their environment and navigate through space. Each grid cell fires when the animal occupies a specific location, independent of movement direction or velocity [1]. When multiple grid cells are active together, they generate a mental map of the surroundings, which is essential for exploring new environments and returning to familiar locations [10,12].

Grid cells function alongside other specialized neurons, such as place cells, head-direction cells, and border cells, which are illustrated in Figure 2. Place cells, discovered in the hippocampus by O’Keefe and Dostrovsky in 1971, fire when the animal is in a particular location [9]. Head-direction cells, located in the entorhinal cortex and postsubiculum, fire when the animal faces a specific direction [8]. Border cells activate near the edges or boundaries of the environment. Together, these cells contribute to a comprehensive spatial map. Many grid cells are modulated by head direction, refining spatial representation, particularly in regions like the presubiculum and parasubiculum where head-direction cells are abundant [13]. The colocalization of grid cells, head-direction cells, and border cells in these regions supports the brain’s ability to continuously update its spatial representation as the environment changes [13].

The grid cell network is essential for path integration—a process by which an animal tracks its movements to estimate its current position, even without visual landmarks. This estimation is achieved through the integration of sensory inputs from the visual, vestibular, and motor systems, allowing the animal to navigate using self-motion cues alone [14,15]. Path integration enables accurate navigation in dynamic environments and supports the formation of mental maps of space, key to spatial cognition and memory [16].

Damage to the MEC or selective silencing of grid cells leads to significant impairments in spatial navigation and memory. Studies have shown that rodents with disrupted grid cell function struggle with object location memory and exhibit disorientation, highlighting the importance of grid cells in maintaining a metric representation of space [1].

## 3. Grid Cell-Based Navigation in Dynamic Environments

Navigating dynamic environments, where obstacles or objects may move unpredictably, poses significant challenges for robots. One approach inspired by grid cells is path integration, which tracks the robot’s movements to estimate its position over time [14]. However, grid cell-based navigation alone may struggle when the environment undergoes substantial changes, as path integration can accumulate errors without external reference points [17]. To address this limitation, grid cell models are often combined with additional strategies such as dynamic mapping. In dynamic mapping, the robot builds and updates a map of the environment in real time, allowing it to adapt to changes like new obstacles or moving objects and to replan its path accordingly [7].

Utilizing grid cell-like representations enhances navigation efficiency by enabling the robot to plan its path through an internal map rather than relying solely on reactive sensor inputs. This capability supports faster and more accurate movement through complex environments. Moreover, grid cell-based navigation provides versatility by allowing robots to adapt to various environments. Unlike traditional navigation algorithms tailored to specific settings, grid cell-based approaches enable robots to generalize and navigate new surroundings without extensive modifications or specialized algorithms, facilitating exploration in unfamiliar territories [5].

## 4. Grid Cells in Robot Navigation

One approach to incorporating grid cell-like representations into robot navigation is the RatSLAM algorithm [7], which combines sensory inputs, such as visual and auditory cues, with a virtual environmental map to create a representation of the robot’s surroundings. This map enables the generation of motor commands to guide the robot through the environment. In RatSLAM, grid cells represent the robot’s position, and as the robot moves, the grid cell network updates its firing patterns to reflect its new location, providing a stable positional reference even when sensory data is limited or noisy.

Interest in grid cell-based navigation for mobile robots has been growing, with models incorporating grid cells into robotic control systems to enhance navigation efficiency and robustness [18,19,20,21]. Key challenges in this area include developing biologically accurate yet computationally efficient models of grid cell firing. Proposed models range from computational ones using circular attractors [14,22] to multi-scale grid configurations [23,24]. Recently, deep neural networks have been used to improve grid cell firing accuracy and efficiency [5,25,26], providing potential advancements in grid cell-based navigation. There are used also recurrent neural network models incorporated to model grid cell behavior [16].

Another challenge lies in integrating grid cell firing with other sensory data, such as vision or sonar [27]. One promising approach is to use a Kalman filter [28] to combine the grid cell model output with sensory input, yielding a more accurate position and orientation estimate. Combining grid cells with place cells, which respond to specific locations, can also create a more precise environmental map [5].

A related challenge is using the grid cell map to plan optimal robot trajectories. Reinforcement learning can associate specific grid cell firing patterns with successful navigation outcomes, allowing the robot to learn flexible and adaptive navigation strategies [5]. This approach enables the development of navigation systems that are both efficient and capable of adapting to dynamic, unfamiliar environments.

## 5. Supervised Learning Grid Cell Module

In our study, we utilized RNN models to develop a Supervised Learning Grid Cell Module (SLGCM) to evaluate the emergence of grid cells in supervised learning for vector-based navigation as illustrated in Figure 3. Current navigation models generally assume that a grid cell network in the entorhinal cortex encodes information about an agent’s speed and movement direction [5]. Our RNN predicts future positions and relays this information to hippocampal place cells, which fire at specific locations, with information flowing bidirectionally between grid and place cells. Evidence indicates that place cells tend to develop prior to grid cells; deactivation of place cells disrupts the formation of the characteristic hexagonal patterns in grid cell activity [12]. This suggests that grid cell processing may depend on positional information from place cells rather than functioning independently. In fact, inactivation of place cells results in the disappearance of grid cell activity, highlighting their dependency on place cell activity, which emerges first [29].

In the simulation, an agent navigated a two-dimensional (2D) space. To train grid cells, trajectory point inputs were generated from the mobile unmanned ground vehicle (UGV) Husky robot in the Gazebo simulation environment. A Long Short-Term Memory (LSTM)-based [30] neural network with dual outputs was implemented to learn from these trajectory points. This strategy leverages a recurrent neural network (RNN) to capture temporal dependencies in grid cell firing patterns. By training the RNN on these sequences, it can predict future firing patterns using learned information from past occurrences, effectively modeling the navigation behavior of grid cells.

### 5.1. Ground Truth Place Cell Distribution

The ground truth place cell distribution represents the inherent spatial arrangement of place cell firing patterns in a given environment. These place cell activations are synthetically generated based on the posterior probability of each element within a combination of two-dimensional isotropic Gaussian functions. The activations represent the neural firing activity of specific neurons in the Supervised Learning Grid Cell Module (SLGCM), and they demonstrate selective firing when a robot occupies specific locations or “places” within the environment. This distribution characterizes the precise locations in the environment where these cells are most likely to exhibit activity.

The ground truth place cell distribution refers to how place cells are ideally distributed across different spatial locations. Place cells exhibit selective firing patterns, becoming active or “firing” in response to specific locations. Each place cell has a preferred firing location or “place field” where it is most active, meaning that when a robot enters that specific location, the corresponding place cell fires with higher intensity. These activations reflect the likelihood of place cells being active in response to a particular spatial location or context. In an ideal situation, where all conditions are known and controlled, place cell activations would precisely follow this distribution. However, real-world factors such as noise or external variations can cause deviations from this ideal distribution.

Place cell activations, represented as $\vec{pcact}$, are vectors with *N* elements, each ranging from 0 to 1, representing normalized activation levels of place cells. A higher activation level means stronger firing activity, indicating the place cell is more responsive to a specific spatial location. The total number of place cells is denoted by *N*. Place cell activations provide numerical values for the neural response of each place cell at specific locations, and this vector representation helps quantify place cell activity in an organized way.

The activations are simulated using the posterior probabilities from a mixture model described by Equation (Equation 1). This model combines multiple two-dimensional isotropic Gaussian distributions to represent the overall distribution of place cell activations. Each Gaussian, defined by its mean and standard deviation, captures the spread and central tendency of activation values. The circular symmetry of the isotropic Gaussians implies that the probability distribution is rotationally symmetric around its center, meaning the likelihood of activations is equally distributed in all directions from the center.

This distribution assigns probabilities to different points in the two-dimensional space, indicating the likelihood of activations in specific regions [31]. The activations, denoted $pcact_{i}$, are calculated using the following formula:(1)$$pcact_{i}=\frac{e^{-\frac{{\left||\vec{posxy}-\vec{pccent_{i}}|\right|}_{2}^{2}}{2{\left(pcscale\right)}^{2}}}}{\sum_{j=1}^{N}e^{-\frac{{\left||\vec{posxy}-\vec{pccent_{j}}|\right|}_{2}^{2}}{2{\left(pcscale\right)}^{2}}}}.$$

Equation (Equation 1) calculates the likelihood that position $\vec{posxy}$ belongs to component *i*, based on the Euclidean distance between $\vec{posxy}$ and the center $\vec{pccent_{i}}$ of component *i*. The center $\vec{pccent_{i}}$ is the mean of the component’s probability distribution, representing the specific location associated with that component. The numerator exponentiates the negative squared Euclidean distance, and the denominator sums the exponentiated distances of all components to normalize the result.

The activations $pcact_{i}$ represent the firing activity of place cells, where each activation value $pcact_{i}$ corresponds to a specific place cell. The calculation of $pcact_{i}$ is based on the position vector $\vec{posxy}$, the mean vector $\vec{pccent_{i}}$, and the scale parameter pcscale. The position vector $\vec{posxy}$ represents the two-dimensional spatial location, while the mean vector $\vec{pccent_{i}}$ denotes the center of each place cell. Equation (Equation 1) normalizes activations by dividing each exponential value by the sum of all exponentiated components, ensuring the activations remain within the range of 0 to 1 and represent relative probabilities.

In Equation (Equation 1), the numerator calculates the weight of a specific component *i* based on the position $\vec{posxy}$. The negative squared Euclidean distance, ${\left||\vec{posxy}-\vec{pccent_{i}}|\right|}_{2}^{2}$, measures the distance between position $\vec{posxy}$ and the center $\vec{pccent_{i}}$. The negative sign in the exponent indicates an inverse relationship between distance and likelihood. Exponentiating the negative squared distance assigns greater weight to components closer to the position $\vec{posxy}$. The denominator in Equation (Equation 1) sums the exponentiated distances of all components *j* from 1 to *N*, where *N* is the total number of components, normalizing the weights to ensure they form a valid probability distribution.

Initialization of place cell centers $\vec{pccent_{i}}$ in the model is important as these centers represent the preferred firing locations of the place cells. The centers, which are the means of each component’s probability distribution in the mixture model, determine the spatial locations associated with the place cells. There are *N* two-dimensional vectors, where *N* is the total number of place cells or components in the mixture model. Each vector $\vec{pccent_{i}}$ represents the center of a specific place cell. During initialization, the centers are generated randomly from a uniform distribution, providing a diverse starting point for learning spatial patterns. The centers define the preferred firing locations of the place cells, establishing a correspondence between each place cell and its associated spatial region.

The parameter pcscale is a positive scalar fixed for each experiment and represents the spread of the Gaussian distributions for each place cell. This scale influences how broadly the activations of place cells extend in the environment. The scale parameter determines the size of the influence of each place cell, and its value remains constant throughout the experiment to maintain consistency. The consistent value of pcscale ensures that all place cells operate with the same spatial influence during the learning process.

This ground truth distribution provides a baseline for evaluating the emergence of place cell-like activity in the SLGCM. While real-world factors such as noise or environmental variations can cause deviations from this ideal distribution, modeling the place cell activations in this manner allows for controlled assessment of the neural network’s ability to learn spatial representations. The synthetic generation of activations facilitates precise characterization of the locations where these cells are most likely to exhibit activity.

### 5.2. Ground Truth Head-Direction Cell Distribution

The ground truth head-direction cell distribution represents the activation pattern of head-direction cells in response to specific facing angles. Head-direction cell activations, denoted as $\vec{hdcact}$, are expressed as a vector of *M* elements, each normalized between 0 and 1, where *M* represents the number of head-direction cells. Each element in $\vec{hdcact}$ indicates the firing rate of a specific head-direction cell, with higher values corresponding to stronger firing rates, thus enabling quantitative assessment of neural activity patterns in head-direction cells.

The activations $hdcact_{i}$ for each head-direction cell are computed using a posterior probability based on a mixture of Von Mises distributions, as shown in Equation (Equation 2):(2)$$hdcact_{i}=\frac{e^{hdconc\;\cos\left(\varphi -hdccent_{i}\right)}}{\sum_{j=1}^{M}e^{hdconc\;\cos\left(\varphi -hdccent_{j}\right)}}.$$

Here, the posterior probability reflects the likelihood of activating each head-direction cell given the observed facing angle φ. This probability is dependent on how each head-direction cell aligns with the robot’s orientation, showing the cells’ responsiveness to specific angular orientations.

The Von Mises distribution, which is circular, models the probability of each head-direction cell firing in response to angles. In this mixture model, each distribution corresponds to a head-direction cell, with activations associated with the posterior probability of each component, representing the likelihood that a given cell is activated based on the facing angle [31]. By combining multiple Von Mises distributions, this model captures the likelihood of the robot facing a particular direction, with head-direction cells selectively firing based on specific orientations, thus supporting spatial orientation and navigation tasks.

The concentration parameter hdconc in the Von Mises distribution controls the distribution’s precision, affecting how tightly the distribution is centered around its mean. A higher hdconc value results in a more concentrated distribution, indicating that head-direction cells have a sharper, more focused preference for a particular direction. This parameter is fixed throughout the experiment to ensure uniform precision in the firing preferences of the head-direction cells.

The head-direction centers $hdccent_{i}$ represent the preferred facing angles for each head-direction cell, meaning each head-direction cell is tuned to respond most strongly to a specific facing angle. The head-direction center indicates the angle at which the cell is most sensitive and exhibits the highest firing rate. During the initialization phase before training, these facing angles are randomly assigned to each head-direction cell. The centers are selected from a uniform distribution, ensuring that any angle within the specified range has equal probability of being assigned. Random initialization at the beginning of training ensures that the head-direction cells do not have any bias or prior knowledge about particular directions. This randomness allows the model to discover optimal tuning patterns for head-direction cells based on the observed data, ensuring diversity in the head-direction representations.

### 5.3. Grid Cell Network Input Data in a Supervised Setup

In a supervised learning context, the grid cell network model as illustrated in Figure 4 is trained on labeled data, where each input is associated with a target value. At each time step *t*, the network receives inputs and adjusts its internal weights based on the difference between the predicted output and the actual labeled target, progressively enhancing predictive accuracy.

The primary input at each time step *t* is the egocentric linear velocity $v_{t}$, which is measured relative to the robot’s own position and orientation, thus defining a frame of reference centered on the observer [32,33]. This egocentric velocity represents the magnitude of the robot’s movement within its frame of reference, providing a scalar measure of linear velocity without accounting for direction.

In addition to $v_{t}$, the network receives the sine and cosine values of the angular velocity $\varphi_{t}$, which represents the rate of change of the robot’s heading direction. This angular velocity data captures the robot’s rotational movement, encoding information about its orientation changes over time. Specifically, the sine of $\varphi_{t}$ reflects the perpendicular component relative to the x-axis, while the cosine of $\varphi_{t}$ reflects the parallel component. These two orthogonal values, taken together, represent the robot’s rotational state at each time step and allow the network to detect angular patterns in the environment by incorporating changes in heading direction.

### 5.4. Computational Model Architecture of Grid Cell Neural Network

The grid cell network is a computational model designed to emulate the spatial navigation behavior observed in animal grid cells, providing a framework for robot navigation. The network consists of multiple interconnected grid cells that collectively encode spatial information by processing inputs, specifically the egocentric linear velocity and the sine and cosine of the angular velocity, to model spatial orientation and movement.

The network architecture, illustrated in Figure 5, comprises three main layers: a recurrent layer, a linear layer, and an output layer. Each layer contributes to transforming and processing input data to produce the desired output. The recurrent layer uses Long Short-Term Memory (LSTM) cells, known for their ability to capture and maintain temporal dependencies in sequential data. LSTM cells address the vanishing gradient problem, enabling the network to retain relevant information over extended sequences, which is essential for tasks such as trajectory prediction. At each time step, the LSTM layer receives the input vector $\left[v_{t},\sin\left(\varphi_{t}\right),\cos\left(\varphi_{t}\right)\right]$, composed of the egocentric linear velocity $v_{t}$ and the sine and cosine values of angular velocity $\varphi_{t}$. This input provides the network with data on both movement and rotation, enabling it to track spatial orientation over time.

The LSTM layer contains a single LSTM cell with 128 hidden units, which maintain and update 128 internal states. These hidden states represent intermediate representations of the input data, capturing temporal dependencies on multiple scales, from short-term to long-term dependencies across distant time steps. Each hidden unit corresponds to a specific feature within the sequence, collectively forming the memory capacity of the LSTM, thereby retaining essential information throughout the process.

Following the recurrent layer, a linear layer transforms the output from the LSTM into the final output layer, which includes place units and head-direction units. This linear transformation applies weight multiplication and bias addition to map the LSTM activations into a suitable output format. The linear layer facilitates the integration of spatial and directional information, supporting the network’s development of an internal spatial representation of the environment. Dropout regularization is applied during training in this layer to prevent overfitting, which enhances the network’s generalization capabilities. By projecting the transformed information to place units, the network can associate specific activation patterns with distinct spatial locations, effectively building a map-like representation of the environment. Similarly, projection to head-direction units enables the model to associate activations with different facing angles, supporting orientation learning.

We selected an LSTM hidden size of 128 as it offers a good balance between representation capacity and training stability. Our primary objective was to demonstrate the concept of grid cell development while minimizing training time and computational demands, rather than comparing different architecture sizes.

A hidden state dimension of 128 for the LSTM and 256 for the subsequent linear layer ensured efficient training without exceeding memory limits. Additionally, many RNN-based models in the literature (excluding extremely large-scale tasks) commonly adopt hidden sizes in the 128–256 range, making our choice of 128 consistent with established practices for moderate-scale sequence modeling.

The integration of temporal data through the LSTM, spatial and directional information through the linear layer, and the combined use of place and head-direction units enable the grid cell network to encode spatial information based on the robot’s trajectory and orientation. This configuration allows the network to model spatially informed decision-making, enhancing the robot’s navigation ability and supporting its operation within a spatial context.

### 5.5. Initialization of LSTM Cell State and Hidden State

The initialization of the LSTM cell state and hidden state provides the LSTM cell with an initial memory configuration that reflects the robot’s spatial location and heading direction. This setup aligns the LSTM’s initial conditions with the context of the input data, enabling the network to effectively capture temporal dependencies and the dynamics of the robot’s movement and orientation.

As defined in Equation (Equation 3), the initial cell state $\vec{Ctinit_{0}}$ is initialized through a linear transformation of the ground truth place cell activations $\vec{pcact_{0}}$ and head-direction cell activations $\vec{hdcact_{0}}$ at time 0. Specifically, this transformation involves multiplying $\vec{pcact_{0}}$ by a weight matrix $W_{Ctpc}$ and $\vec{hdcact_{0}}$ by another weight matrix $W_{Cthdc}$:(3)$$\vec{Ctinit_{0}}=W_{Ctpc}*\vec{pcact_{0}}+W_{Cthdc}*\vec{hdcact_{0}}.$$

The transformed vectors encapsulate the contributions of the place and head-direction cells, and an element-wise summation operation integrates these into a single vector, producing the initial cell state $\vec{Ctinit_{0}}$.

Similarly, as shown in Equation (Equation 4), the initial hidden state $\vec{htinit_{0}}$ is computed by applying a linear transformation on the same ground truth vectors $\vec{pcact_{0}}$ and $\vec{hdcact_{0}}$:(4)$$\vec{htinit_{0}}=W_{htpc}*\vec{pcact_{0}}+W_{hthdc}*\vec{hdcact_{0}}.$$

In this case, separate weight matrices $W_{htpc}$ and $W_{hthdc}$ are used to integrate the influence of the place and head-direction cells into the hidden state, ensuring that both cell types contribute distinctively to the initial hidden state.

The matrices $W_{Ctpc}$, $W_{Cthdc}$, $W_{htpc}$, and $W_{hthdc}$ are learnable parameters that are optimized during training. Through iterative gradient-based optimization, the network adjusts these matrices to minimize the difference between predicted outputs and ground truth values, refining the initialization for both cell and hidden states. These matrices form essential parameters within the network, continually updated during training to enable accurate and effective initialization of the LSTM’s states, ultimately enhancing predictive performance.

### 5.6. Linear Decoder Architecture

The linear decoder network transforms the high-dimensional output state from the LSTM into predictions for the activities of place cells and head-direction cells. Through supervised training on labeled data, the decoder learns to adjust its weights and biases to minimize the error between predicted and actual activations, refining its capacity to model spatial and directional representations accurately.

During sequence processing, the LSTM cell performs computations on sequential data, updating its cell and hidden states based on the current input and prior states. At each time step *t*, the LSTM produces an output state $\vec{ht_{t}}$, encapsulating the learned representations up to that point. This output state $\vec{ht_{t}}$ serves as a condensed form of the input sequence, retaining relevant patterns identified by the LSTM, and it provides the basis for the decoder’s subsequent predictions.

The linear decoder network subsequently transforms $\vec{ht_{t}}$ into predicted values for place cells $\vec{pcpred_{t}}$ and head-direction cells $\vec{hdcpred_{t}}$. The decoder’s weight matrices act as learnable parameters capturing relationships between the LSTM’s output and the target predictions. These weights are adjusted iteratively to align the predictions with ground truth values optimally. Notably, the linear decoder operates without non-linear activation functions, relying on straightforward linear transformations.

Place cell predictions allow estimation of the robot’s location, as these cells correspond to specific spatial coordinates. Analyzing these predictions enables inference of the robot’s probable position over time, assisting in tracking spatial navigation. Similarly, predictions for head-direction cells provide estimates of the robot’s orientation, allowing interpretation of its perception of environmental direction and orientation-based behavior. Combining predictions from place cells and head-direction cells yields a more complete understanding of the robot’s spatial state, which may be further refined by integrating additional sensory data or computational models.

The linear decoder is structured with three sets of weights and biases. The first set maps the LSTM’s hidden state $\vec{ht_{t}}$ to linear layer activations $\vec{linlact_{t}}$, a vector of 256 elements, transforming the hidden state into a format suitable for final predictions.

The second and third sets of weights and biases map $\vec{linlact_{t}}$ to predicted place cells $\vec{pcpred_{t}}$ and head-direction cells $\vec{hdcpred_{t}}$. Softmax functions are applied to these mappings to convert activations into probability distributions over potential head directions and place cell states, ensuring normalized and valid probabilities. These softmax-transformed values represent the likelihood of the robot’s head direction and spatial position.

Dropout with a probability of 0.5 is applied to each unit in the linear layer activations $\vec{linlact_{t}}$.

To improve prediction accuracy, the decoder adjusts transformation parameters, minimizing discrepancies between $\vec{pcpred_{t}}$ and $\vec{hdcpred_{t}}$ and their corresponding ground truth values, thus continually refining the mappings to enhance performance in representing spatial and directional information.

## 6. Simulated UGV Robot Environment

The simulated environment shown in Figure 6 or the Husky Rover is set up in Gazebo using the Robot Operating System (ROS), creating a controlled space for training spatial navigation tasks. In this environment, the robot navigates within a square area enclosed by wooden walls, which serve as boundaries and resemble the setup of the Morris Water Maze. Unlike the water boundaries in the original Morris Water Maze, these walls provide a physical constraint that limits the robot’s movement to a defined area, guiding its exploration within a compact vista space.

Vista space, as classified by Montello [34], refers to a spatial extent that can be perceived from a single position or with minimal movement. This concept is important for the training setup, as it emphasizes spatial cues that are immediately visible to the robot without requiring extensive exploration. The enclosed environment in Figure 6 provides a well-defined perimeter for the robot’s navigation tasks, supporting structured learning within an accessible spatial range. This setup enables the robot to develop spatial awareness in a bounded environment, allowing it to focus on immediate spatial information for effective navigation training.

## 7. Mobile UGV Robot Trajectory Dataset Generator Module (RTDGM)

The Mobile UGV Robot Trajectory Dataset Generator Module (RTDGM) interacts with the Simulated UGV Robot Environment Interface Module (EIM) to produce trajectory data for the Husky robot within a Gazebo simulation environment as illustrated in Figure 7. As seen in the provided diagram, this process starts with the generation of core components, including egocentric velocity, position coordinates, and head direction. These elements, created by respective generators—egocentric_velocity_generator, pos_coord_generator, and head_direct_coord_generator—supply the necessary inputs for guiding the simulated robot’s trajectory.

The EIM module facilitates communication between the dataset generator and the simulation environment by accepting commands compatible with the OpenAI Gym interface [35], such as env.reset(), env.step(action), and env.render(). These commands are converted into Gazebo-specific instructions, enabling the control of the simulated UGV robot as well as environmental elements like walls and obstacles. When the env.step() function is called, the EIM module provides the dataset generator with values for next.obs, rewards, and dones, which are essential for generating comprehensive trajectory data.

In the simulation setup, the Husky robot’s trajectory is generated within a square enclosure with wall lengths of L=6.4 m. In Figure 8, each trajectory spans 100 steps, with the robot beginning from a random position and orientation within the central 2.1×2.1 m area. The robot’s movements are governed by a discrete action space comprising seven possible actions, allowing fine-grained control over its rotation from −2.25 rad to 2.25 rad in increments of 0.75 rad. A random motion model ensures that the robot explores the environment uniformly, avoiding wall collisions [36], and moves continuously forward while receiving movement commands.

The dataset generation process also incorporates a coordinate frame transformation to convert local orientations within the Gazebo environment to the global orientation framework required by the Supervised Learning Grid Cell Module (SLGCM). The resulting dataset is stored in 100 TFRecord files, with 70 files designated for training and 30 for evaluation, making it compatible with machine learning models for spatial and directional learning.

As illustrated in Figure 9, the Trajectory generator module produces control commands (sent to husky_velocity_controller/cmd_vel) that guide the simulated Husky robot’s motion. These commands are processed through ROS and utilize the husky_robot ROS package, which provides necessary launch files, scripts, and hardware drivers. The diagram further shows that the Gazebo simulation environment supports both virtual and real Clearpath Husky robots, allowing for seamless transitions between simulated and real-world testing.

The Husky robot’s odometry information is obtained via ROS topics (ground_truth/state with message type nav_msgs/Odometry), providing real-time feedback on the robot’s position and orientation. This feedback is processed by the dataset generator to refine trajectory data, ensuring alignment with the global orientation required for the SLGCM framework.

## 8. Materials and Methods

The training and simulation environment for this project was implemented using the Gazebo simulator (version 11.9.0, Open Robotics, Mountain View, CA, USA) integrated with the Robot Operating System (ROS) Noetic Ninjemys (Open Robotics, Mountain View, CA, USA) to model the Clearpath Husky UGV robot (Clearpath Robotics Inc., Kitchener, ON, Canada). A trajectory dataset generator module was designed to create simulated navigation trajectories based on randomized starting locations and orientations within a 6.4 m × 6.4 m square enclosure. The environment, controlled through the Environment Interface Module (EIM), utilizes commands compatible with the OpenAI Gym API (version 0.26.2, OpenAI, San Francisco, CA, USA) [35] to manage robot actions and obtain observations, rewards, and completed states, which were essential for generating data aligned with reinforcement learning protocols.

The EIM translates high-level actions into Gazebo-specific commands, interfacing with ROS to control the Husky robot’s movement using the husky_velocity_controller/cmd_vel topic. Trajectories were generated in sequences of 100 steps, with the robot executing one of seven discrete actions to adjust its rotation from −2.25 rad to 2.25 rad in increments of 0.75 rad, while continuously moving forward to cover the environment uniformly [36]. Coordinate frame transformations were applied to map local orientations in Gazebo to a global orientation used in the Supervised Learning Grid Cell Module (SLGCM).

Training data were stored in TFRecord files, with 70 files allocated for training and 30 for evaluation. This data format, compatible with TensorFlow (version 2.9.1, Google, Mountain View, CA, USA), facilitates efficient loading and processing during model training. The generated dataset captures both positional and directional information by incorporating activations from place cells and head direction cells to represent the robot’s spatial state.

The training was executed on a system equipped with an AMD Ryzen 5 3600 6-Core Processor (Advanced Micro Devices, Inc., Santa Clara, CA, USA), 32 GB RAM, and an NVIDIA RTX3060 12 GB GPU (NVIDIA, Santa Clara, CA, USA), running Ubuntu 20.04 LTS (Canonical, London, UK). CUDA 11.4 (NVIDIA, Santa Clara, CA, USA) was used to accelerate computation on the GPU. TensorFlow 2.9.1 provided the machine learning framework, supporting the training and evaluation of the model within a neural network architecture consisting of LSTM layers and linear decoders to simulate grid cell and head direction cell activations.

## 9. Results

The grid cell network was trained using simulated trajectories generated by the Husky UGV robot (manufactured by Clearpath Robotics) in a bounded environment. The primary goal of the experiment was to assess the emergence of spatially tuned responses, including grid-like, border-like, and head-direction-like activations, within the neural network’s linear layer. After 124 training epochs, distinct spatial activity patterns emerged in the linear layer of the network. To visualize these responses, we generatedratemaps by plotting each unit’s activation intensity over the two-dimensional environment. The ratemaps were computed by binning the robot’s navigational space into a regular grid and averaging the activation levels of each unit per spatial bin. Figure 10 displays the spatial activity ratemaps of various linear layer units.

Each subfigure corresponds to a unit, with activation intensities color-coded (low: blue; high: red). Units exhibiting periodic, hexagonally arranged firing fields are indicative of grid-like encoding. The ratemaps exhibit periodic and spatially distinct activation regions across the environment, demonstrating hexagonal symmetry consistent with biological grid cells. Notably, units with higher activation values exhibit clearer grid-like patterns, indicating stronger spatial tuning. To further analyze these patterns, Figure 11 presents spatial autocorrelograms derived from the ratemaps in Figure 10. The autocorrelograms were derived by computing Pearson correlation coefficients between the ratemap and its shifted versions.

The autocorrelograms in Figure 11 highlight the radial symmetry and periodicity of the learned activations, confirming the network’s ability to develop hexagonal spatial representations. These results strongly suggest that the grid cell network successfully encodes spatial relationships in a biologically plausible manner.

To quantitatively assess the spatial encoding, we calculated the gridness score, a metric to evaluate hexagonal regularity in grid-like patterns. Briefly, the gridness score compares the two-dimensional spatial autocorrelogram of a unit’s firing pattern to rotated versions of itself at 30°, 60°, 90°, 120°, and 150°. Higher gridness scores indicate greater resemblance to a hexagonal grid arrangement. The top-performing units in the linear layer achieved gridness scores between 0.65 and 0.73 (values shown in parentheses in Figure 10 and Figure 11). Units with higher gridness scores exhibited more defined periodic structures and hexagonal symmetry, as evident in their autocorrelograms.

The learned representations in the linear layer demonstrate a biologically inspired hierarchy of spatial encodings, including grid-like units that exhibit hexagonal periodicity and symmetry, resembling biological grid cells; border-like units with activation patterns concentrated near the environment boundaries, analogous to border cells; and head-direction-like units that display tuning to specific orientations, similar to head-direction cells. The diversity of these representations highlights the network’s ability to generalize spatial information, a critical feature for navigation tasks. This diversity aligns with known functional heterogeneity in the mammalian navigational system and indicates that the artificial neural network can learn rich spatial representations from derived trajectory data.

These results demonstrate that an artificial network, when provided with suitable trajectories and environmental constraints, can develop representations that mirror key properties of biological spatial cells. These results suggest that the grid cell network effectively learned spatial representation by leveraging the generated trajectories of the Husky robot. The trained network’s ability to mimic grid and directionally tuned cells provides a robust foundation for further exploration in robotic spatial navigation applications, using artificial neural networks to model spatial cognition.

## 10. Discussion

Despite advancements in grid cell-based navigation in robotics, several challenges remain. One major challenge is developing algorithms that can efficiently handle large-scale environments, where computational complexity and resource demands increase significantly. Another is achieving better integration with various sensory modalities, including vision and auditory systems, which are essential for creating a comprehensive spatial understanding. Additionally, more realistic simulations are needed to test navigation systems under conditions that closely resemble real-world environments and more dynamic trace data of robot movement trajectories are needed to further test grid-cell algorithm navigational ability in real world environments. Also, performance analyses are needed to the estimate metrics of accuracy of the navigation of the grid cell-based algorithm.

Future research could benefit from developing more biologically inspired models of grid cells. Current models primarily focus on the hexagonal grid patterns observed in the medial entorhinal cortex, yet other grid-like patterns may exist in different brain regions. Investigating these patterns could lead to models that more accurately reflect the biological mechanisms of grid cell firing, potentially improving navigation algorithms.

The proposed grid cell–inspired approach can serve as an internal path integration mechanism, providing a self-motion–based estimation of position and heading without external references. However, much like biological systems, purely grid cell–based path integration can accumulate drift over time. Consequently, in many real-world scenarios, it is more practical to integrate this approach with other navigation methods (e.g., SLAM, visual odometry, inertial odometry) to regularly correct any accumulated errors.

Deep learning techniques, particularly convolutional neural networks (CNNs), offer additional avenues for acquiring grid cell representations for navigation. By training a CNN on datasets of robot trajectories, the network can learn to capture spatial firing patterns that mimic grid cell activity. These acquired representations could be highly valuable for navigation, especially in environments with complex spatial structures.

Another area worth exploring is the use of spatial neural representations in three-dimensional navigation algorithms, such as NeuroSLAM [37]. This could expand the applicability of grid cell-based navigation from 2D to 3D spaces, providing more versatility for robots operating in varied environments.

By pursuing these research directions, it may be possible to create more robust, adaptable, and biologically accurate grid cell-based navigation systems, with applications extending from simple 2D navigation to complex, multi-sensory 3D spatial tasks.

## 11. Conclusions

Here is a summary of the key scientific contributions that differentiate our approach:Realistic, physics-based training data. Most previous work on grid cell–inspired robotics relies on purely synthetic or simplified 2D simulations without realistic physics. We developed a pipeline to collect training data from a ROS + Gazebo environment, including actual dynamics (collisions, motor delays, sensor noise) that closely approximate real robots. This allows the learned grid-cell representations to be more readily transferable to physical robots.Emergence of multiple cell types. Although many studies focus solely on grid cells in simulated mazes, our network simultaneously exhibits grid cell–like, border cell–like, and head-direction cell–like activation patterns, reflecting a broader biological realism. We demonstrate how these different spatial representations arise within the same architecture trained on realistic robot trajectories.Bridge from neuroscience to practical robotics. While many grid-cell models remain conceptual or limited to small-scale lab simulations, our method demonstrates how to implement and train such a model for a mobile robot in a realistic environment, thereby advancing potential real-world applications (e.g., path integration and state estimation).

Taken together, these contributions represent a step forward in bridging neuroscience-inspired navigation strategies and robotic implementations.

Despite the progress, challenges remain. Efficient algorithms for large-scale environments, better integration with sensory modalities like vision and auditory systems, and the development of more realistic simulation environments are necessary to advance grid cell-based navigation in robotics further. Addressing these challenges will enable robots to navigate with greater adaptability and resilience, especially in environments with dynamic obstacles and varying sensory inputs.

The potential applications of grid cell-inspired navigation systems are vast, ranging from search and rescue operations to autonomous transportation. By applying biologically-inspired spatial and directional representation, robots can achieve improved localization, orientation, and adaptability, even in unfamiliar or unpredictable environments. Future research directions could include developing more biologically accurate grid cell models, exploring hybrid models that integrate grid cells with cognitive processes such as memory and decision-making, and applying these models in three-dimensional navigation systems such as NeuroSLAM [37].

In conclusion, using grid cell principles in robotics presents a promising path forward for creating robust and adaptive navigation systems. Continued research and development in this field can bring robotic navigation closer to human-like spatial reasoning, ultimately enabling robots to navigate with precision, adaptability, and resilience in a wide range of real-world scenarios.

## Figures and Tables

**Figure 1 sensors-25-01576-f001:**
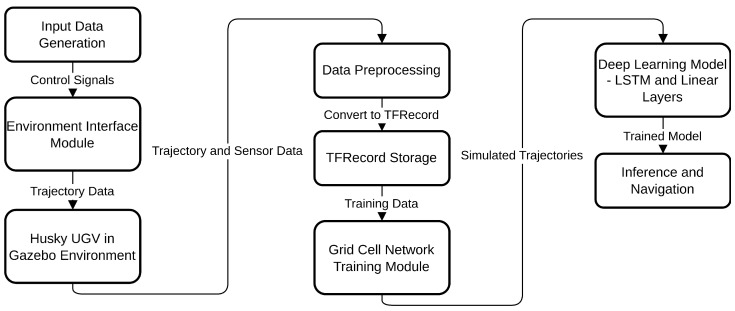
An overview of the high-level system architecture for grid cell-based robotic navigation, detailing the flow from input data generation through trajectory simulation in Gazebo, data preprocessing and storage, model training with Long Short-Term Memory (LSTM) and linear layers.

**Figure 2 sensors-25-01576-f002:**
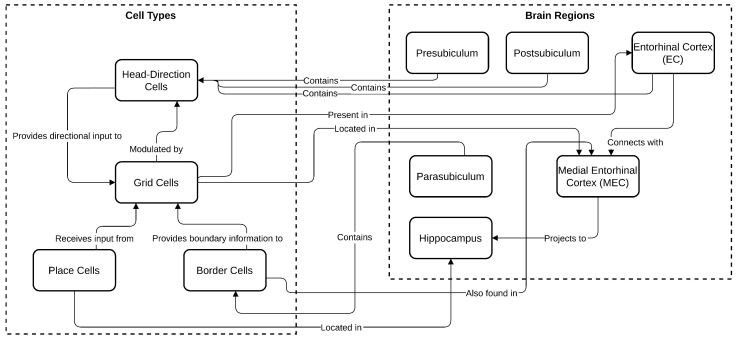
The diagram illustrates the relationships between various brain regions and cell types involved in spatial navigation.

**Figure 3 sensors-25-01576-f003:**
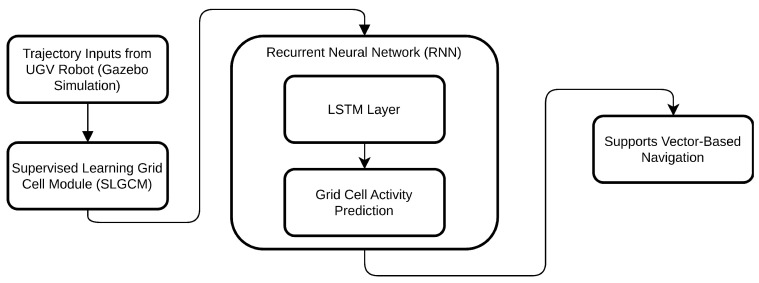
The diagram illustrates the flow of data from trajectory inputs of a UGV robot in a simulation environment through the Supervised Learning Grid Cell Module (SLGCM), using an RNN and LSTM layer to predict and simulate grid cell activity that supports vector-based navigation.

**Figure 4 sensors-25-01576-f004:**
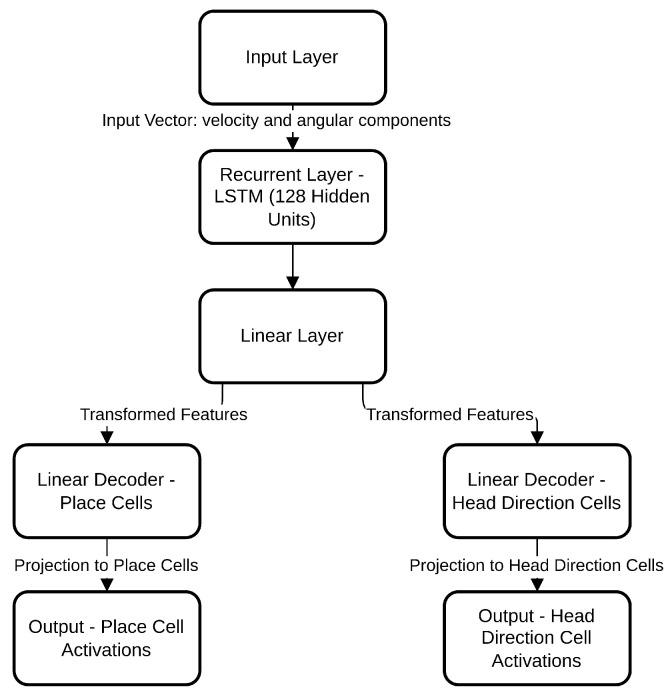
Architecture of a grid cell network for robotic navigation, detailing the flow from the input layer (taking velocity and angular components) through the LSTM recurrent layer, a linear layer for feature transformation, and separate linear decoders for place and head-direction cells, which output activations for spatial and directional mapping.

**Figure 5 sensors-25-01576-f005:**
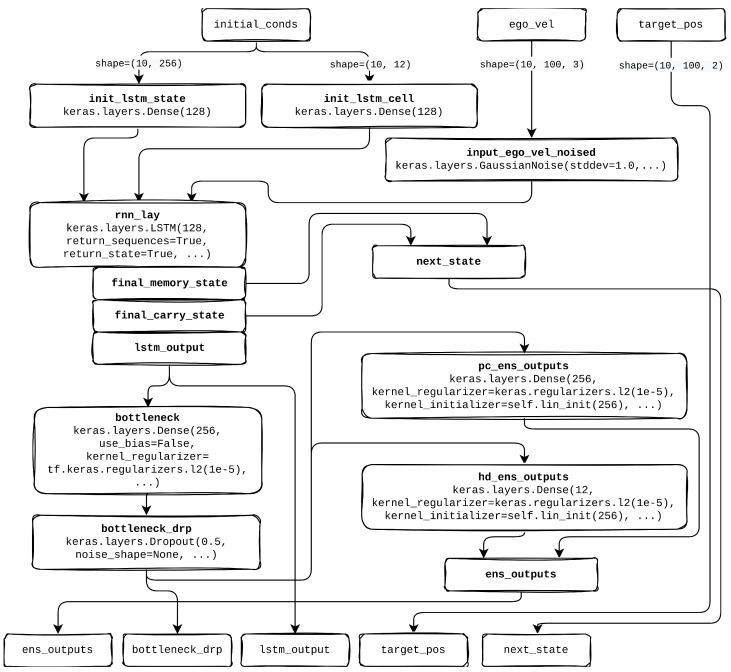
Tensorflow 2.9.1 implementation architecture of supervised learning grid cell module.

**Figure 6 sensors-25-01576-f006:**
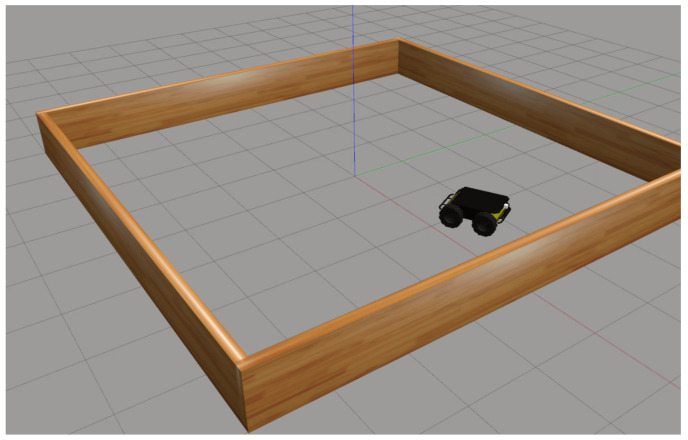
Husky UGV robot in Gazebo environment.

**Figure 7 sensors-25-01576-f007:**
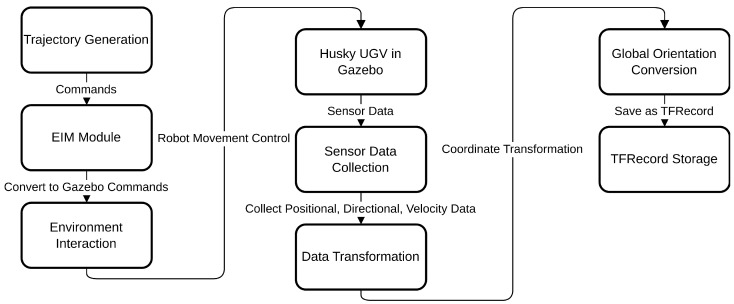
The flow of dataset generation and environment interaction for the Husky UGV in a Gazebo simulation, showing how commands are processed through the EIM module, robot movements are controlled, sensor data are collected and transformed, and the data is finally stored in TFRecord format for training purposes.

**Figure 8 sensors-25-01576-f008:**
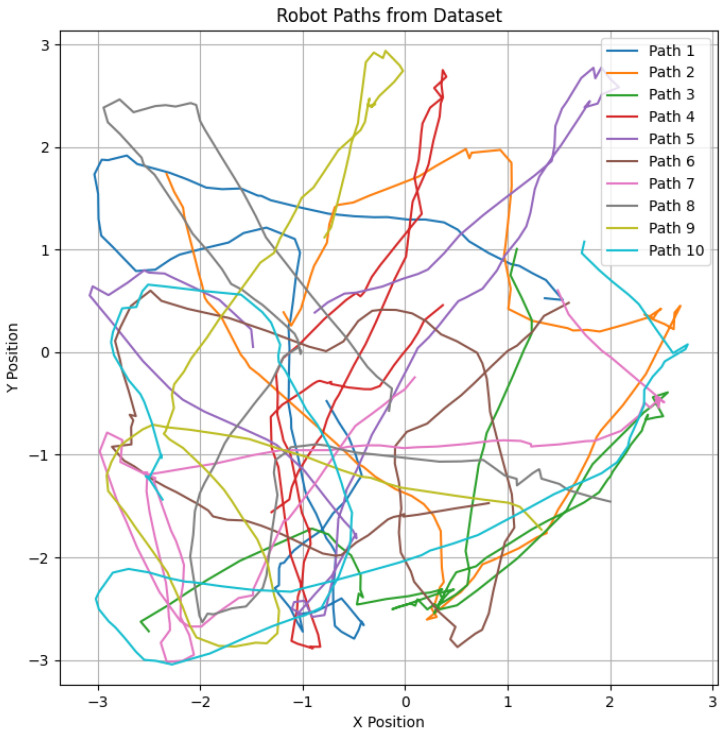
Husky robot’s traveled trajectories during dataset generation, with each colored line representing a different path within a 6.4-meter-square area (±3.2 m on both the X and Y axes).

**Figure 9 sensors-25-01576-f009:**
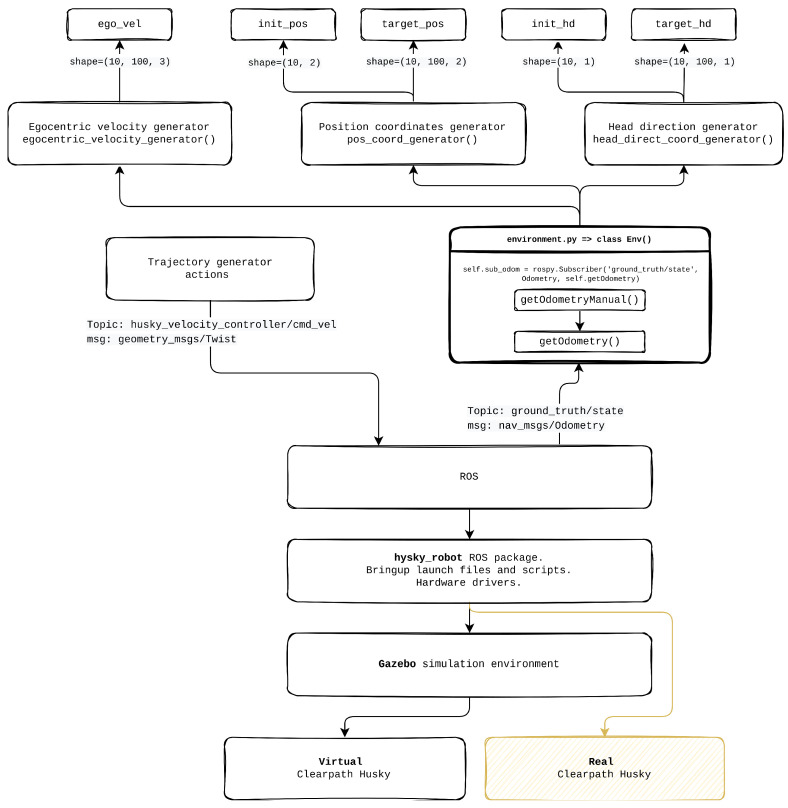
Dataset trajectory generation from Clearpath Husky in Gazebo to dataset in TFRecords format.

**Figure 10 sensors-25-01576-f010:**
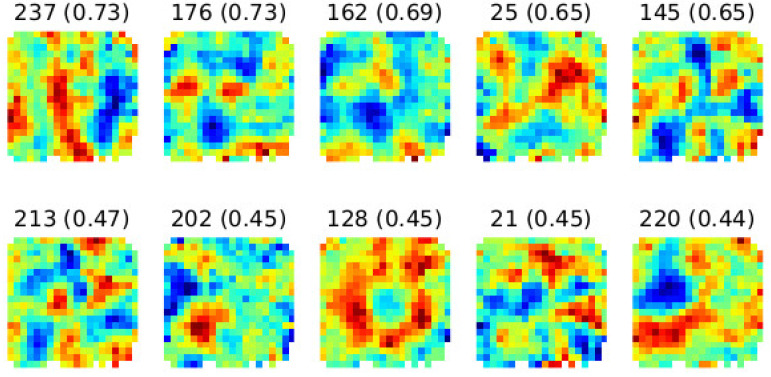
Spatial activity plots from the linear layer after 124 training epochs show similar to grid cell activations. These activations exhibit periodic and varied firing patterns, reflecting learned spatial encoding across different units.

**Figure 11 sensors-25-01576-f011:**
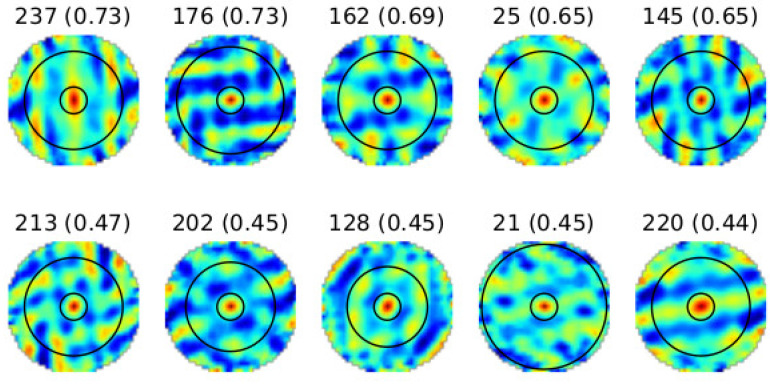
Spatial autocorrelograms of ratemaps after 124 training epochs, displaying distinct circular patterns that reflect spatial regularity.

## Data Availability

The datasets presented in this article are not readily available because the data are part of an ongoing study and file size limitations. Requests to access the datasets should be directed to corresponding author.

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
