# Peer review of "Deep Learning-Emerged Grid Cells-Based Bio-Inspired Navigation in Robotics"

_sensors, 2025, doi:10.3390/s25051576_

Round 1

Reviewer 1 Report

Comments and Suggestions for Authors

(1)The sampling data set of trace for Supervised learning is simple and less. Can you provide more complicated simulation environment in Gazebo simulation to collect more dynamic trace data. And give the performance analysis of your navigation grid cell network.

 (2)The performance analysis “Accuracy of navigation” of the grid cell network should be added.  

Reviewer 2 Report

Comments and Suggestions for Authors

The main idea of ​​the work is very interesting and promising, it is related to the use of the latest results of biological research on rats in the navigation of autonomous robots for their orientation in space. The topic of supporting navigation tools is really very relevant and the direction studied in the work is of interest.

The authors have provided a good, concise introduction that describes the bio-inspired approach of grid cells.
But I could not catch what exactly new is proposed by the authors that allowed to make the technology "more adaptive and robust"? This part needs more explanation. I had a hope that in other parts of the paper I would meet smth but even in conclusion I have not seen this.

In the descriptional part of the grid technology there are too many unneeded biological terms even in those parts where more formal tecnical terms are expected (for exapmle in lines 128-133). Why animal speed if a robot is considered as an object?

It is better to add some part with explanation of abbreviations. For example, in Fig.1 LSTM is used, but the explanation appeares only at 143 line.

As a strong point of the work, I would note a detailed description of the functionality of the entire technology, including various types of cells, as well as a detailed and clear description of the neural network implementation.

But there are some questionary points to be clarified.

- Lines 359-364. If linear decoder operates without non linear activation functions, so previous linear transformations can be united with decoding linear transformations into one?

- It would be interesting to understand whether this approach can be used as an independent navigation mechanism? or only as a support system for the existing SLAM, visual or inertial odometry? what are the limitations of its applicability in this case?

- In the presented neural network implementation, the authors chose the layer dimensions as 128 and 512, respectively. It would be worthwhile to provide a justification for this choice. Perhaps the results of experiments with other parameters.

- And most importantly, even in the conclusion I was unable to understand what exactly (except for the detailed software implementation) the authors managed to do new? What was proposed from a scientific point of view? What new was achieved?

Now in the conclusion the advantages of the already proposed approach are presented. But what is the scientific novelty of this particular work remained unclear to me.

Reviewer 3 Report

Comments and Suggestions for Authors

This paper provides an overview of recent research on grid cell-based navigation in robotics for uncertainty and dynamic environments and experimental results where a grid cell network was trained using trajectories from a mobile unmanned ground vehicle robot. The reviewer has the following comments:

1. From a review perspective of recent research, more literature should be included.

2. The advantages of the proposed grid cell neural network compared to alternative network architectures are not clearly demonstrated. Including a comparative analysis with other state-of-the-art methods would be beneficial.

3. While the authors point out that existing studies have primarily focused on simulated environments, this paper also relies on trajectory simulation in Gazebo for input data generation. Real-world experimental validation should be provided to demonstrate practical applicability.

4. Additional details regarding training efficiency metrics and performance benchmarks would be beneficial.

5. The Gazebo environment used is relatively simplistic. Testing in more complex and varied environment configurations would help demonstrate the robustness of the method.

Round 2

Reviewer 1 Report

Comments and Suggestions for Authors

This method is totally new thread of navigation. The authors can give the testing in real robot and real world to prove it further.  

Reviewer 3 Report

Comments and Suggestions for Authors

The revised version is not significantly improved compared to the previous version. In my opinion, the strength of the contribution remains weak compared to references in the field, but the proposed work may be of interest.